# Can the Synergic Contribution of Multigenic Variants Explain the Clinical and Cellular Phenotypes of a Neurodevelopmental Disorder?

**DOI:** 10.3390/genes13010078

**Published:** 2021-12-28

**Authors:** Nuno Maia, Maria João Nabais Sá, Cláudia Oliveira, Flávia Santos, Célia Azevedo Soares, Catarina Prior, Nataliya Tkachenko, Rosário Santos, Arjan P. M. de Brouwer, Ariana Jacome, Beatriz Porto, Paula Jorge

**Affiliations:** 1Unidade de Genética Molecular, Centro de Genética Médica Jacinto de Magalhães (CGM), Centro Hospitalar Universitário do Porto (CHUPorto), 4099-028 Porto, Portugal; nuno.maia@chporto.min-saude.pt (N.M.); fa_by@live.com.pt (F.S.); rosariosantos.cgm@chporto.min-saude.pt (R.S.); 2Unit for Multidisciplinary Research in Biomedicine (UMIB), Laboratory for Integrative and Translational Research in Population Health (ITR), Institute of Biomedical Sciences Abel Salazar (ICBAS), University of Porto, 4050-313 Porto, Portugal; maria.nabaissa@gmail.com (M.J.N.S.); celiasoares@chporto.min-saude.pt (C.A.S.); 3Laboratório Citogenética, Instituto de Ciências Biomédicas Abel Salazar (ICBAS), Universidade do Porto, 4050-313 Porto, Portugal; csoliveira@icbas.up.pt (C.O.); bporto@icbas.up.pt (B.P.); 4Serviço de Genética Médica, Centro de Genética Médica Doutor Jacinto Magalhães (CGM), Centro Hospitalar Universitário do Porto (CHUPorto), 4099-028 Porto, Portugal; natalia.tkachenko@chporto.min-saude.pt; 5Unidade de Neurodesenvolvimento do Serviço de Pediatria do Centro Materno-Infantil do Norte (CMIN), Centro Hospitalar Universitário do Porto (CHUPorto), 4050-651 Porto, Portugal; catarinaprior.cmin@chporto.min-saude.pt; 6Department of Human Genetics, Donders Institute for Brain, Cognition and Behaviour, Radboud University Nijmegen, 6525 GA Nijmegen, The Netherlands; arjan.debrouwer@radboudumc.nl; 7Chromosome Instability and Dynamics Lab. (CID), Instituto de Inovação e Investigação (i3S), Universidade do Porto, 4200-135 Porto, Portugal; ariana.jacome@ibmc.up.pt

**Keywords:** neurodevelopmental disorder, chromosome instability, DNA repair pathways

## Abstract

We describe an infant female with a syndromic neurodevelopmental clinical phenotype and increased chromosome instability as cellular phenotype. Genotype characterization revealed heterozygous variants in genes directly or indirectly linked to DNA repair: a de novo X-linked *HDAC8* pathogenic variant, a paternally inherited *FANCG* pathogenic variant and a maternally inherited *BRCA2* variant of uncertain significance. The full spectrum of the phenotype cannot be explained by any of the heterozygous variants on their own; thus, a synergic contribution is proposed. Complementation studies showed that the *FANCG* gene from the Fanconi Anaemia/BRCA (FA/BRCA) DNA repair pathway was impaired, indicating that the variant in *FANCG* contributes to the cellular phenotype. The patient’s chromosome instability represents the first report where heterozygous variant(s) in the FA/BRCA pathway are implicated in the cellular phenotype. We propose that a multigenic contribution of heterozygous variants in *HDAC8* and the FA/BRCA pathway might have a role in the phenotype of this neurodevelopmental disorder. The importance of these findings may have repercussion in the clinical management of other cases with a similar synergic contribution of heterozygous variants, allowing the establishment of new genotype–phenotype correlations and motivating the biochemical study of the underlying mechanisms.

## 1. Introduction

Neurodevelopmental disorders (NDDs) are defined as “a group of conditions with onset in the developmental period […] characterized by developmental deficits that produce impairments of personal, social, academic, or occupational functioning” [1,2]. They represent an important health burden, affecting more than 3% of children worldwide [3]. Up to 60% of individuals affected with NDDs are carriers of de novo (likely) pathogenic genetic variants [1]; nevertheless, the majority of NDDs remain molecularly undiagnosed. Massive parallel sequencing, commonly named next-generation sequencing (NGS), became a vital methodology in the establishment of the molecular diagnosis, with especial impact in diseases with high clinical and genetic heterogeneity or nonspecific phenotypes [4]. One such example is the rare inherited chromosome breakage syndrome, Fanconi anaemia (FA), characterized by multiple congenital abnormalities, haematological defects and predisposition to malignancies [5,6]. FA patient cells show an exceptional sensitivity to DNA cross-linker agents such as mitomycin C (MMC) and dexproxibutane (DEB) [7]; therefore, the presence of increased chromosome breakage and radial forms on cytogenetic testing allows the establishment of FA diagnosis. This is also attained when one of the following is identified upon molecular genetic testing: biallelic pathogenic variants in one of the 21 autosomal recessive FA genes, a heterozygous variant in *RAD51* gene or a hemizygous pathogenic variant in the X-linked *FANCB* gene. FA/BRCA pathway proteins are mostly involved in DNA repair in a genome maintenance process, essential to DNA-DNA cross-link and DNA double-stand breaks repair mechanisms [8]. In this process, the FA core complex, composed by FANCA, -B, -C, -E, -F, -G, -L, -M, is responsible for the monoubiquitination of FANCD2 and FANCI, and the remainder, namely FANCD1 (BRCA2), -J (BRIP1), -N (PALB2), -O (RAD51C), -P (SLX4) and -Q (XPF), act downstream of this step [9].

NGS also has advanced the identification of multiple variants in more than one gene, with a combined/cumulative effect. Clinical manifestations resulting from variants in different genes are increasingly being reported [10,11]. In the digenic diseases database (DIDA, Available online: http://dida.ibsquare.be (accessed on 13 December 2021) at least 54 diseases, caused by 258 digenic combinations, involving 169 genes and 448 pathogenic variants, are described [12]. However, experimentally evaluating the combination of these variants for deleteriousness and causality is currently unworkable given the large amount of data generated and the number of variants identified in each individual genome.

Herein, we describe an infant female affected with NDD, which we hypothesise resulted from a synergic contribution of variants in the genes directly or indirectly linked to DNA repair. This also represents the first report where heterozygous variant(s) in FA/BRCA repair pathway gene(s) are implicated in an increased chromosome instability.

## 2. Materials and Methods

### 2.1. Subjects

Peripheral blood samples from the proband, both parents, a healthy donor (negative control) and Fanconi anaemia (FA) patients (positive controls) were used in this study. Confirmation of the FA diagnosis was performed at the Laboratory of Cytogenetics, ICBAS, UP. Epstein–Barr virus-transformed lymphoblast cell lines (LCLs) were generated from peripheral blood cells of the proband and a negative control. As positive controls, LCLs from three FA patients were used. FA LCLs were kindly provided by Dr Juan Bueren’s Laboratory at the Centro Investigaciones Energéticas, Medioambientales y Tecnológicas (CIEMAT), Madrid. Primary fibroblasts from the proband were also used in this study.

All procedures were carried out with the informed consent of the participants. This investigation was approved by the Ethical Committee on Human Research of Centro Hospitalar Universitário do Porto (CHUPorto.)—REF 2015.196 (168-DEFI/157-CES) and Instituto de Ciências Biomédicas Abel Salazar (ICBAS), Universidade do Porto (UP)—PROJETO No. 129/2015.

### 2.2. Cell Cultures

Peripheral blood was cultured, for 72 h, in RPMI-1640 (Sigma-Aldrich, St. Louis, MO, USA), at 37 °C in a 5% CO_2_, humidified atmosphere, supplemented with 15% foetal bovine serum (FBS; Sigma-Aldrich), 1% of penicillin/streptomycin (Pen/Strep; Lonza, Basel, Switzerland), 29 mg/mL of L-glutamine (Sigma-Aldrich) and 5 μg/mL of phytohemagglutinin (Gibco, Thermo Fisher Scientific, Waltham, MA, USA), for stimulation of lymphocytes.

Epstein–Barr virus-transformed lymphoblast cell lines (LCLs) were cultured in RPMI-1640 (Sigma-Aldrich) supplemented with 15% FBS (Gibco, Thermo Fisher Scientific) and 1% of Pen/Strep (Lonza) at 37 °C in a 5% CO_2_, humidified atmosphere.

The proband’s primary fibroblasts were cultured in Dulbecco’s Modified Eagle’s Medium (DMEM; Gibco, Thermo Fisher Scientific) supplemented with 15% FBS (Gibco, Thermo Fisher Scientific) and 1% Pen/Strep (Lonza) at 37 °C in a 5% CO_2_, humidified atmosphere, until 90–95% confluence.

### 2.3. Molecular Studies

#### 2.3.1. Exome Sequencing, Variant Filtering and Prioritization, and CNV Calling

Exome Sequencing (ES) analysis was performed on proband’s genomic DNA (gDNA), obtained from peripheral blood, using salting out methods [13]. Libraries were captured using a SureSelect V5-post Kit (Agilent Technologies, Santa Clara, CA, USA), and 100 bp paired-end sequencing was performed using an Illumina HiSeq 2000/2500 (Illumina, San Diego, CA, USA). The Genome Analysis Toolkit (GATK v3.4.0) was used to assemble the raw data of FASTQ file format into the University of California Santa Cruz (UCSC) Genome Browser (Available online: http://genome.uscs.edu/ (accessed on 10 February 2016))—human assembly: February 2009 (hg19—NCBI build GRCh37) and variant alleles were annotated using SnpEff (SnpEff_v4.1 g).

Variants passing following filters were selected for clinical correlation: (a) frequency < 1% (dbSNP, GnomAD Browser, and local databases); (b) gene component, that is, exon and canonical splice acceptor or donor sites; (c) non-synonymous consequence; (d) in silico deleteriousness and spliceogenic effect predictions, using tools: (i) Combined Annotation Dependent Depletion scoring (CADD threshold ≥ 15) [14]; (ii) SpliceSiteFinder-like (SSF, normal score threshold ≥ 70 for SDS and SAS) [15]; (iii) MaxEntScan (MES, normal score threshold ≥ 0 for SDS and SAS) [16]; (iv) NNSPLICE (NNS, normal score threshold ≥ 0.4 for SDS and SAS) [17]; and (v) GeneSplicer (GS, normal score threshold ≥ 0 for SDS and SAS) [18]. Variants are described according to Human Genome Variation Society (HGVS) recommendations [19,20], and classification of clinical significance follows the guidelines of the American College of Medical Genetics and Genomics and the Association for Molecular Pathology (ACMG and AMP) [21] (Varsome, Available online: https://varsome.com (accessed on 28 October 2021) [22] and ClinVar classification [23].

To confirm the presence of the selected variants and further segregation studies, Sanger sequencing was performed using the primers: (i) gDNA_HDAC8-F 5′-CACTACCCCTAGACCAAACTGACC-3′ and gDNA_HDAC8-R 5′-AAAGACACTTGCCAATTCCCAC-3′; (ii) gDNA_FANCG-F 5′-CTCGAGGCACCTGAAGTAGG-3′ and gDNA_FANCG-R 5′-GCTTCTCTGCAATGGGGTAG-3′; (iii) gDNA_BRCA2-F 5′-TGATCCACTATTTGGGGATTG-3′ and gDNA_BRCA2-R 5′-TCTCTGGACCTCCCAAAAAC-3′. PCR products were purified using the Illustra^TM^ ExoStar^TM^ 1-Step, (GE Healthcare Life Sciences, Little Chalfont, UK), sequenced using the BigDye Terminator v3.1 cycle sequencing kit (Applied Biosystems, Foster City, CA, USA) and further analysed with SeqScape Software v2.5 (Applied Biosystems).

CNV calling based on ES data was performed using CoNIFER (Available online: http://conifer.sourceforge.net/ (accessed on 28 May 2019)) [24]. CNVs with an absolute Z-score greater than 1.7 were considered for analysis. All deletions were considered disruptive, as were duplications in known fully penetrant microdeletion/duplication regions and intragenic CNV duplications. CNVs that overlapped known regions of partial penetrance were considered separately [25].

#### 2.3.2. Transcript Analysis in Proband’s Blood and Fibroblasts

Blood and fibroblast RNA samples were obtained from proband and controls, using 5 PRIME PerfectPure RNA Blood and Tissue Kits, respectively (Thermo Fisher Scientific, Waltham, MA, USA). *HDAC8* exon 8 cDNA was amplified using SuperScript One-Step—RT-PCR with Platinum Taq kit (Invitrogen, Carlsbad, CA, USA) following the manufacturer’s instructions, using primers cDNA_HDAC8-F 5′-CAGGTGACGTGTCTGATGTTG-3′ and cDNA_HDAC8-R 5′-ACCCCGGTCAAGTATGTCC-3′. PCR products were purified using Illustra ExoStar 1-Step, (GE Healthcare Life Sciences), followed by an asymmetric PCR using a BigDye Terminator v3.1 cycle sequencing kit (Applied Biosystems).

### 2.4. Test for Chromosome Instability Evaluation

The DEB test was used for confirmation/exclusion of FA diagnosis. The standard protocol for the DEB test has been well established at the Laboratory of Cytogenetics, ICBAS, UP. In summary, lymphocyte cultures were treated with DEB (Sigma-Aldrich) at 0.05 μg/mL and 0.1 μg/mL during 48 h at 37 °C in a 5% CO_2_ humidified atmosphere. Metaphase arrest was obtained after a treatment with colcemid (Gibco, Thermo Fisher Scientific) for 1 h at 37 °C in a 5% CO_2_ in a humidified atmosphere, following hypotonic solution (KCl, 0.75 M) (Merck, Kenilworth, NJ, USA) treatment. After fixation with a 3:1 ice-cold solution of methanol (Thermo Fisher Scientific) and acetic acid (Thermo Fisher Scientific), the cell suspension was dropped onto a properly-labelled microscope slide. Slides were left to air dry for approximately 24 h (room temperature and 50–60% humidity) and stained with 4% Giemsa (Merck) for 4 min. Using the Olympus CX31 microscope, metapahases were selected with a 100X objective lens and captured using an Olympus EP50 camera.

For each lymphocyte culture, 100 metaphases were analysed in a blinded fashion. Each cell was scored for chromosome number, and the number and the types of structural abnormalities: breaks (achromatic areas wider than a chromatid), fragments (also scored as breaks), dicentric chromosomes, ring chromosomes and chromatid exchange configurations (triradial and tetraradial figures), the last three being scored as rearrangements and considered as two breaks. The chromosome instability parameters evaluated were percentage of aberrant cells and mean number of breaks per cell. Reference values for FA classification were established according to Auerbach et al. [26]. The mean number of breaks per cell, in cultures with DEB concentration of 0.05 μg/mL, is the diagnostic discriminative parameter, because there is no overlap of values between AF (DEB positive) and non-AF/control (DEB negative) groups (Table 1).

### 2.5. Mitomycin C Sensitility Test

To challenge the FA/BRCA pathway, LCLs were submitted to increased concentrations of MMC (0–1000 nM, M0503, Sigma-Aldrich) in fresh medium for a period of 120 h. After this period, cells were resuspended in phophate-buffered saline (PBS)–bovine serum albumin (BSA; 0.05%) containing 0.5 mg/mL propidium iodide (P-4864, Sigma-Aldrich) and incubated for 10 min at 4 °C. Cell viability was determined by flow cytometry based on the PI exclusion test. The analysis was carried out, taking into account the viability of the 0–3 nM cells as a reference in each cell type condition. All the doublets were discarded from the analysis. Flow cytometry analysis was performed on an FACS Calibur (Becton-Dickinson, San Jose, USA). As internal controls, a healthy donor (normal control) was used in each assay, along with three FA patients (positive controls in one of the experiments (kindly provided by Dr Juan Bueren´s laboratory).

### 2.6. Complementation with Retroviral Vectors Expressing Functional FANCG

To investigate whether FANCG-expressing vectors could restore the phenotype of proband’s cells, LCLs were transduced with vectors expressing FANCG protein (LGEG11) [27,28]. In all instances, vectors were packaged in PG13 cells, and titers of 0.5–5 × 10^6^ infective particles/mL were routinely obtained. To transduce LCLs, cell culture plates were pre-treated in a plasma chamber (Diener Electronics, Ebhausen, Germany) for 2 min and subsequently coated with fibronectin (2 mg/cm^2^, reference F1141, Sigma-Aldrich). Wells were preloaded twice with 1 mL of retroviral supernatants for 30 min at 37 °C. After virus preloading, 3 × 10^5^ cells in 2 mL of supplemented RPMI were seeded to the wells. After three weeks, sufficient cells were obtained in all the conditions, and the mitomycin C (MMC, 0–1000 nM) sensibility test was performed.

## 3. Results

### 3.1. Clinical Case

A six-month-old girl, the second child of a non-consanguineous couple, was referred for genetic consultation due to global developmental delay, Pierre-Robbin sequence and failure to thrive. Intrauterine growth restriction was noticed at 20 weeks of gestation. She was born by caesarean section at 38 weeks with an Apgar index of 7/8, weight and length below the 5th percentile and an occipital frontal circumference (OFC) at the 5th percentile. At 8.5 months of age, her weight, height and OFC were below the 5th percentile. She had a global developmental delay, including absent speech. At physical examination, she had craniofacial dysmorphisms (Figure 1), including lacrimal duct obstruction and cleft palate; digital abnormalities including short 5th metacarpal, low-set thumb, broad hallux, hypoplastic nails of the fifth toes; limited hips abduction; scoliosis; hirsutism; and hypopigmentation spots. Additionally, axial hypotonia with peripheral hypertonia was identified. Gastro-esophageal reflux was also diagnosed. At the age of 6.8 years, she has low weight (−5.7 SD), short stature (−3.8 SD) and microcephaly (−5.6 SD). After intensive speech and language therapy, she uses sign language to communicate.

Bilateral hearing loss was confirmed by auditory evoked potentials (bilateral electrophysiological limits of 80 dB). An echocardiogram, performed at four months of age, showed patent foramen ovale and persistent left superior vena cava. Brain MRI at the age of three months revealed (i) large sylvian fissures, especially on the left, suggesting delayed “operculation”; (ii) a retrocerebelar paramedian right space dilation, likely corresponding to an arachnoid cyst; (iii) a shaped, although apparently well-formed, inferior side of the cerebellar vermis; (iv) prominent cortical sulci; (v) a thin corpus callosum; (vi) a less prominent brain stem; (vii) persistence of small germinolytic cysts; and (viii) an adequate myelination pattern for that age. She presented recurrent haematological anomalies, such as monocytosis and thrombocytosis.

Initial genetic and metabolic investigations were normal:important karyotype, fluorescence in situ hybridisation (FISH) of 22q11.2, microarray-based comparative genomic hybridisation (aCGH), *NIPBL* gene sequencing, new-born metabolic screening [29] and reducing sugars in urine. Furthermore, TORCH congenital infections were excluded.

### 3.2. Exome Sequencing Analysis

Heterozygous variants in genes associated with known autosomal dominant NDDs were first prioritized due to the high incidence rate of de novo variants in Western countries and the absence of consanguinity in this family. However, this approach failed to identify strong candidate variants as well as relevant CNVs. A promising heterozyguos candidate variant was identified in the *HDAC8* gene (Xq13.1; OMIM*300269), NM_018486.2:c.793G>A, p.(Gly265Arg), which, after confirmation by Sanger sequencing and segregation studies, proved to be de novo (Figure 2).

Transcript analysis revealed differences between different tissues: in blood, the normal allele was expressed exclusively, whereas in cultured fibroblasts a weak expression of the mutant allele was observed (Figure 3).

Unavailability of other tissues, namely brain, hampers the assessment of the likely pathogenic *HDAC8* heterozygous variant involvement in the phenotype. A revision of 45 females affected with *HDAC8* variants was performed (Appendix A). Despite overlap with some of the proband’s clinical features, such as short stature, low weight, psychomotor delay, poor speech, hypotonia, dysmorphic features and other complications including feeding problems, gastroesophageal reflux, cardiovascular defects, hearing loss and lacrimal duct obstruction [30,31,32,33,34,35], the *HDAC8* variant does not fully explain the phenotype of this infant female.

### 3.3. Chromosome Instability Evaluation

The presence of haematological abnormalities, together with mild digital defects, led to chromosome instability studies (DEB test) in order to confirm/exclude an FA cellular phenotype. As shown in Table 2, the diagnostic parameter number of breaks per cell (DEB concentration of 0.05 µg/mL) classifies the proband as DEB negative. Therefore, an FA cellular phenotype was excluded. However, both the number of breaks per cell and the percentage of aberrant cells were above the normal values. In cultures exposed to DEB at the concentration of 0.1 µg/mL, increments 3.5-fold and 2.2-fold were observed in the number of breaks per cell and percentage of aberrant cells, respectively. Additionally, some of the aberrant cells showed multiple chromosome breaks and multiple radial figures (Figure 4), which are not found in DEB-induced cells from normal controls. In the corresponding cultures from the negative control, these increments were not observed. These results suggest that the cells from the proband have a hypersensitivity to the clastogenic effect of DEB. In DEB-induced lymphocyte cultures from both parents, the values of percentage aberrant cells and the mean numbers of breaks per cell, both with 0.05 µg/mL and 0.1 µg/mL concentrations, were within the normal range.

The increased level of chromosome instability in the proband’s cells prompted ES data reanalysis focusing on variants in DNA repair genes.

### 3.4. Mitomycin C Sensitivity Test of Proband’s Cells

A concentration-dependent hypersensitivity to MMC was observed in the proband’s LCLs, with a cellular survival rate around 63.1 % ± 2.2 % (*n* = 5) at MMC 33 nM, in contrast to the healthy donor (normal control), 89.8 % ± 2.1 % (*n* = 5) and three FA patients (positive controls) presenting different hipersensitivities (15.1 %; 52.4 %; 37.7 %) (Figure 5).

### 3.5. Reanalysis of ES Data Focusing on DNA Repair Genes

A heterozygous pathogenic variant NM_004629.1:c.1433+1G>A, p.(?) in the *FANCG* gene, which codes for an FA core complex protein, was identified. Segregation studies revealed that this variant was paternally inherited. A second hetrozygous variant, NM_000059.3:c.8293T>C, p.(Cys2765Arg) in the *BRCA2* gene, maternally inherited with an uncertain clinical significance, was also identified.

### 3.6. Complementation Studies with Retroviral Vectors Expressing Functional FANCG

To assess the involvement of the FA/BRCA pathway in the instability, *FANCG* vectors were used in complementation studies. In the proband’s LCLs, complementation with retroviral vectors expressing functional *FANCG* rescued the cellular phenotype to normal control values (Figure 6).

## 4. Discussion

The proband described herein carries the de novo variant NM_018486.2:c.793G>A, p.(Gly265Arg) in the *HDAC8* gene, with a CADD score of 27.8, not annotated in the gnomAD database and classified as likely pathogenic according ACMG/AMP guidelines (Varsome) and ClinVar (RCV000680270.1). *HDAC8* belongs to the histone deacetylase (HDACs) family of enzymes participating in key biological processes such as gene expression regulation, stress response and DNA repair [36], and it is also implicated in the cohesinopathy Cornelia de Lange Syndrome, type 5 (CdLS-5; OMIM#300882) [31] and Wilson-Turner-like phenotypes [30]. Classically, hemizygous HDAC8 males are more severally affected than females, who show a variable but less severe phenotype [30,31]. This variability is usually assumed to be caused by the presence of distinct patterns of X-chromosome inactivation (XCI), as seen in other X-linked disorders such as Borjeson–Forssman–Lehmann, Christianson, Fragile X and Opitz–Kaveggia syndromes, among others (reviewed by Migeon et al.) [37]. Transcript analysis in the proband’s blood revealed the exclusive expression of the normal allele. Although the possibility of an amplification bias cannot be ruled out, we hypothesise a complete skewing of the XCI pattern, in line with previous studies [31,33,34]. The severe neurodevelopmental syndromic phenotype, together with lack of CdLS-5 pathognomonic facial dysmorphisms, suggest that the *HDAC8* variant alone does not fully explain the phenotype of this infant female, namely the haematologic anomalies (Appendix A).

A differential diagnosis was performed with a DNA repair disorder associated with haematological abnormalities, the well-known FA, where the cellular hypersensitivity to DNA cross-linking agents, such as DEB or MMC, is the hallmark for the diagnosis [26]. In FA cells, defects in DNA double-strand-break repair mechanisms lead to chromosome instability, due to biallelic pathogenic variants in any of the genes from the FA/BRCA pathway [26,38]. DEB sensitivity studies in primary lymphocytes from the proband indicated exclusion of FA, but an increased chromosome instability, compared to the normal control, which prompted ES data reanalysis. The paternal inherited heterozygous *FANCG* variant NM_004629.1:c.1433+1G>A, p.(?) is not described in gnomAD, is classified as pathogenic according ACMG/AMP guidelines (Varsome) and is reported in ClinVar as being of uncertain clinical significance (RCV001194963.1). To the best of our knowledge, an instability cellular phenotype has never been described in carriers of a heterozygous variant in FA/BRCA genes (e.g., parents and brothers of FA patients).

Sensitivity to MMC at the differential concentration of 33 nM allows detection of putative FA patients, where healthy donors present survivals above 75% (the majority higher than 80%) [27]. The proband´s sensitivity to MMC was not as high as that of an FA patient but was far from the healthy donor´s values, corroborating the results obtained with the DEB sensitivity test. To investigate if the heterozygous variant in *FANCG* was involved in this phenotype, FANCG complementation studies were performed. Proband’s cells MMC sensitivity was rescued to values close to those of the healthy donor, indicating a rescue in the proficiency of the FA/BRCA pathway function. Despite the exclusion of FA diagnosis, the proband shows some clinical features overlapping with previously reported patients, such as short stature, low weight, peculiar facies, ear and thumb abnormalities and other complications including feeding problems, cardiovascular defects and hearing loss [39]. Interestingly, a maternally inherited heterozygous *BRCA2* variant NM_000059.3:c.8293T>C, p.(Cys2765Arg) was also identified. This variant is not reported in gnomAD and is described as being of uncertain significance according to ACMG/AMP guidelines (Varsome) and in ClinVar (RCV000524906.4). A putative role of this variant, in synergy with the other two, cannot be discarded.

It is plausible to propose a multigenic synergic contribution of the identified variants towards this unique NDD phenotype. The management of such a complex condition and proper follow up of this patient and her family is reoriented as new symptoms emerge. Knowing that chromosome instability can have consequences in human health, particularly the possible effect on tumour predisposition, we highlight the importance of these studies in cases with heterozygous variants in DNA repair genes, even though a de novo variant in other gene(s) (*HDAC8* as in this clinical case) might be identified. In fact, it is known that FA patients naturally tend to decrease the levels of HDAC8 [40], indicating that both pathways have a natural advantage when downregulated.

## 5. Final Remark

We raised the hypothesis of a possible synergic contribution linking DNA repair variants affecting both the FA/BRCA pathway and HDAC family. This had a major impact in disease prognosis and familial clinical follow-up, with other medical specialities being involved in personalized healthcare and adapted to the new emerging symptoms. The identification of other cases with a synergic contribution of heterozygous variants, probably underdiagnosed, will allow the establishment of new conditions with clinical impact.

## Figures and Tables

**Figure 1 genes-13-00078-f001:**
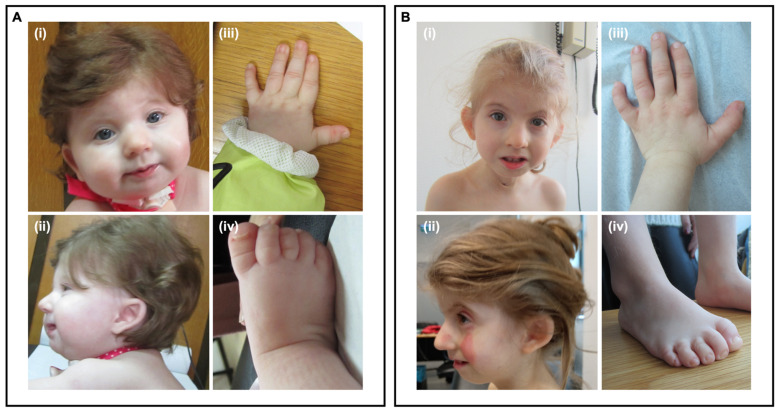
Craniofacial, hand and foot dysmorphisms at nine months old (**A**) and five years and seven months (**B**). (**i**)—Facial dysmorphic features: broad nasal bridge and thin upper lip. (**ii**)—Micrognathia, low-set, posteriorly rotated, dysmorphic ears. (**iii**)—Short 5th metacarpal. (**iv**)—Broad hallux and hypoplastic nails of the fifth toes.

**Figure 2 genes-13-00078-f002:**
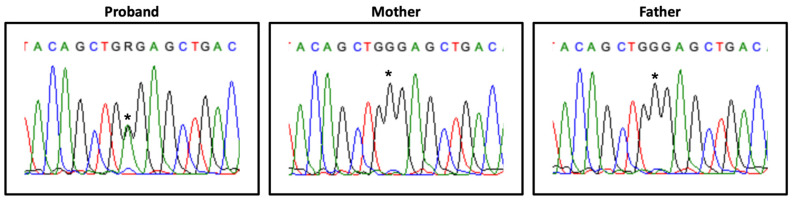
*HDAC8* variant segregation. Partial *HDAC8* exon 8 electropherogram, showing the heterozygous variant NM_018486.2:c.793G>A (*) in the proband and its absence in the parents’ samples.

**Figure 3 genes-13-00078-f003:**
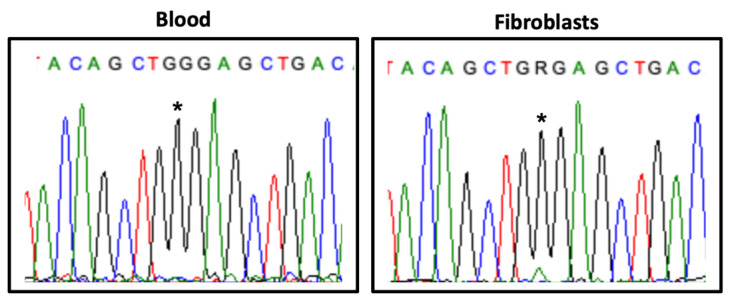
*HDAC8* transcript analysis in blood and fibroblasts. Partial *HDAC8* exon 8 electropherogram showing the expression of the variant NM_018486.2:c.793G>A (*) in the proband’s samples.

**Figure 4 genes-13-00078-f004:**
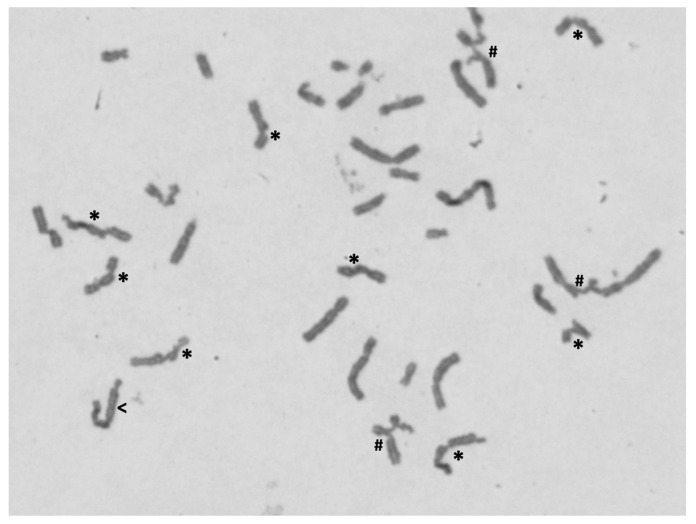
Example of metaphase spread showing chromosome abnormalities. Metaphase from proband’s lymphocyte culture, after treatment with 0.10 µg/mL of DEB, observed under the 100X objective, showing chromosome breaks (*), radial figures (#) and a dicentric chromosome (<).

**Figure 5 genes-13-00078-f005:**
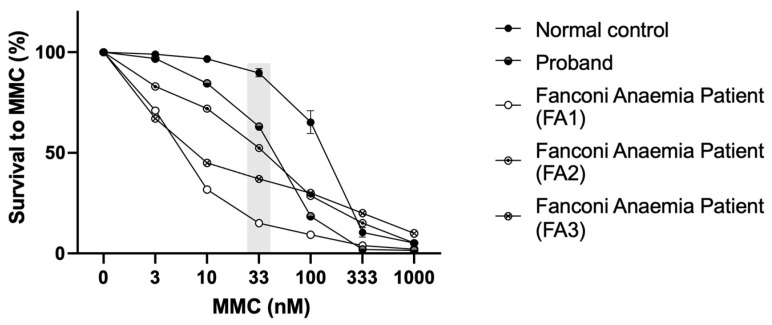
Mitomycin C sensitivity test. MMC sensitivity of the LCLs from the proband and controls (normal control, i.e., healthy donor, and three positive controls, i.e., Fanconi anaemia patients—FA1; FA2; FA3). The differential concentration (33 nM) is highlighted in grey.

**Figure 6 genes-13-00078-f006:**
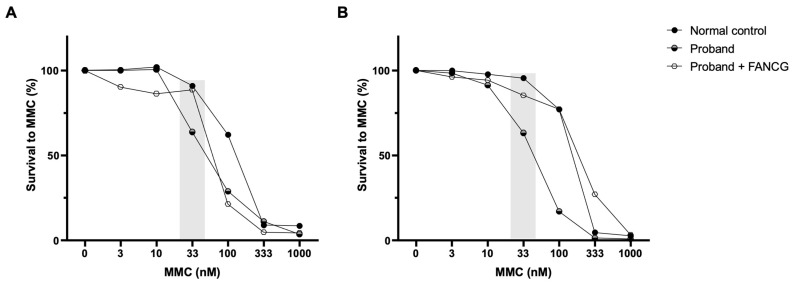
Complementation studies. Genetic complementation of the proband’s LCLs with a retroviral vector expressing a functional FANCG protein (LGEG11). Two different and independent experiments are shown (**A**,**B**). A reversion in the MMC hypersensitivity of the proband’s LCLs after retroviral-mediated gene transfer was observed. The differential concentration (33 nM) is highlighted in grey.

**Table 1 genes-13-00078-t001:** Established reference values for chromosome instability in DEB-treated peripheral blood lymphocytes (adapted from Auerbach et al. [26]).

Parameters	Group	Mean	Min	Max
Nr breaks/cell	FA	8.96	1.30	23.90
	Non-FA/control	0.06	0.00	0.36
% ab cells	FA	85.15	12.60	100.00
	Non-FA/control	5.12	0.00	22.00
Difference between FA and non-FA/control groups for the two parameters: number of breaks per cell (Nr breaks/cell) and percentage of aberrant cell, i.e., cells with breaks (% ab cells).				

Min—minimum; Max—maximum; Nr—number; ab—aberrant; FA—Fanconi anaemia.

**Table 2 genes-13-00078-t002:** Evaluation of DEB-induced chromosome instability (CI) in primary lymphocytes from the proband and parents. Parallel CI evaluations in primary lymphocytes from a healthy donor and a Fanconi anaemia (FA) patient were used as negative and positive controls, respectively. For each sample, evaluation of CI was performed in a total of 100 metaphases. The classification of the proband as FA (DEB positive) or non-FA (DEB negative) was performed according to the reference values indicated in Table 1.

DEB-Induced CI	DEB Concentration: 0.05 µg/mL (Diagnostic Discriminative Parameters to Compare with Reference Values in Table 1)		DEB Concentration 0.1 µg/mL	
	% ab Cells	Nr Breaks/Cell	% ab Cells	Nr Breaks/Cell
Proband	34	0.54	76	1.88
Mother’s Proband	4	0.06	5	0.05
Father’s Proband	1	0.01	3	0.03
Healthy Donor (Negative Control)	3	0.04	2	0.02
FA Patient (Positive Control)	89	5.36	No Metaphases	

DEB—diepoxybutane; CI—chromosome instability; FA—Fanconi anaemia; ab—aberrant; Nr—number.

## Data Availability

The data presented in this study are available in the article and supplementary material.

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
