# Peer review of "Can the Synergic Contribution of Multigenic Variants Explain the Clinical and Cellular Phenotypes of a Neurodevelopmental Disorder?"

_genes, 2021, doi:10.3390/genes13010078_

Round 1

Reviewer 1 Report

Maiaet al., investigated the contribution of multigenic variants in the clinical and cellular phenotypes of a neurodevelopmental disorders.

Although some interesting questions and results are presented in the paper, I have unfortunately identified a number of major concerns including misuse of statistical methodology and crucial misinterpretations of the data. In summary, the most detrimental flaw I find in this manuscript is: The manuscript should be revised and also the writing should be improved with regards to English usage to provide a clearer presentation. The methods section should provide a better description of the nature of the data and the introduction should provide a clearer motivation for the technical challenges being addressed. In addition, what technical/medical/biological insights are gained? Figure captions should be more informative

Please see more my detailed feedback below.

  1. The abstract is also weak and need to be more comprehensive to present the big picture of the study.
  2. The authors should include the methodology they used for variant calling. How did you set the parameters to call variants. Have you found any CNV or chromosome abnormality in the individual genome?

CNVs are the major genetic contributor in the NDDs. Have you look at the CNVs in the genome? Author should look at the CNVs reported in the WES data and find out if there is either rare CNV or previously know CNV in the genome of the affected individuals. There is a comprehensive study of CNVs and the affected genes in paper by Rokny et al, Cell Reports. [PMID: 33113368]. Authors can look at the brain-enriched genes reported in this study to find out if there is any overlap between brain-enriched genes and the affected genes found in their WES data.

  1. I think it would be interesting if author perform a gene ontology analysis to find out if the multigenic variants are involved in brain-related pathways. Also DECIPHER database can be a very useful source to look at the phenotype similarity between the affected indivudlas with previously reported NDD individuals.

  1. Grammatical problems:

There are many grammatical problems in the paper:

For example, I would say neurodevelopmental disorders instead of neurodevelopmental disorder as it refers to more than 100 different disorders.

I never heard de novo disorders! Genetic Disorders might be a better work.

Author Response

ANSWER TO REVIEWER 1

genes-1494316

The comments from Reviewer 1 were greatly appreciated. The authors addressed all the issues raised and prepared a point-by-point cautious consideration. Based reviewer’s queries, the authors consider that the article was greatly improved, as well as the manuscript readability and quality. Changes are marked up using the “track changes” function.

Although some interesting questions and results are presented in the paper, I have unfortunately identified a number of major concerns including misuse of statistical methodology and crucial misinterpretations of the data.

The authors thanks this comment and recognize that there was a misuse of statistical methodology and that the results regarding the chromosomal instability studies could led to misinterpretation(s). Therefore, the authors clarified the methods and introduced “Table 1” (instead of inadequate graphical representation) to facilitate the visualization of the reference values. Also, the results section of the “chromosome instability evaluation” was changed and the values of the experiments are now described in a more comprehensive manner, at the “Table 2”in accordance with FA diagnostic guidelines.

The manuscript should be revised and also the writing should be improved with regards to English usage to provide a clearer presentation.

The manuscript was carefully revised to improve the English, and the authors consider that now it is more readable.

The methods section should provide a better description of the nature of the data and the introduction should provide a clearer motivation for the technical challenges being addressed. In addition, what technical/medical/biological insights are gained?

The methods section was significantly changed, and the authors believe that these sections are now more informative.

1. The abstract is also weak and need to be more comprehensive to present the big picture of the study.

The abstract was rewritten. We trust that it is important for the clinical community that we identified this case. This first observation of a synergism between HDAC8 and Fanconi Anaemia biochemical pathways will be a starting point for several physicians and research groups to revisit their data and work on it. Future research will try to dissect the mechanism underlying the observed synergism between HDAC8, FANCG and BRCA2.

2. The authors should include the methodology they used for variant calling. How did you set the parameters to call variants. Have you found any CNV or chromosome abnormality in the individual genome?

Methodology for variant calling is indicated at methods section. A new paragraph of CNV calling approach was included.

CNVs are the major genetic contributor in the NDDs. Have you look at the CNVs in the genome? Author should look at the CNVs reported in the WES data and find out if there is either rare CNV or previously know CNV in the genome of the affected individuals. There is a comprehensive study of CNVs and the affected genes in paper by Rokny et al, Cell Reports. [PMID: 33113368]. Authors can look at the brain-enriched genes reported in this study to find out if there is any overlap between brain-enriched genes and the affected genes found in their WES data.

The authors recognize the significant implication of CNVs in NDDs. Normal aCGH, and negative CNVs analysis based on exome sequencing data, were obtained. This data is now more clearly indicated both in methods and results sections

4. I think it would be interesting if author perform a gene ontology analysis to find out if the multigenic variants are involved in brain-related pathways.

The authors performed an ontology analysis, revealing an interaction between FANCG and BRCA2 (which is expected as they act together at FA/BRCA pathway) and a genetic interaction between HDAC8 and BRCA2 (please see image below). Furthermore, it is described that FA patients naturally tend to decrease the levels of HDAC8 indicating that both pathways have a natural advantage when downregulated (mentioned in the discussion). The authors believe that the identification of other cases, and future biochemical studies involving these pathways could bring new insights on these ontology interactions. Nevertheless, the authors consider that this analysis is out of the scope of this paper, where the authors intend to show laboratory validated evidences, raising new hypothesis for further studies.

Also DECIPHER database can be a very useful source to look at the phenotype similarity between the affected individuals with previously reported NDD individuals.

The authors did not observe any report case with the same “combination” of affected genes. However, the authors mention at the manuscript the patient shows clinical features similar to previous report HDAC8-females and FA patients.

5. Grammatical problems

The manuscript was carefully revised, and the authors consider that the manuscript is grammatically better.

Thanks to reviewer’s comments the manuscript is now clearer and more understandable.

All authors provided their inputs and agree with this revised version of the manuscript. We trust the current version meets with your approval, with that of the reviewers as well as adhering to the journals’ formatting requirements.

Thanking in advance.

Yours sincerely,

Paula Jorge

Reviewer 2 Report

The authors reported an infant female with a syndromic neurodevelopmental phenotype, increased chromosome instability, presenting heterozygous variants in genes directly or indirectly linked to DNA repair. This is an interesting report.

She had a pathogenic variant of HDAC8 gene. A heterozygous pathogenic variant in FANCG gene was identified. Clinical significance of the variant in the BRCA2 gene was uncertain. They considered that contribution of HDAC8 and FA/BRCA pathway multigenic heterozygous variants might explain the clinical manifestation of this neurodevelopmental disease. It is doubtful that heterozygous pathogenic variant in FANCG causes chromosomal instability. Microdeletion of FANCG may be found by MLPA study. Or RNA seq may find other variants in FANCG or other genes.

Author Response

ANSWER TO REVIEWER 2

genes-1494316

The authors addressed all the issues raised and prepared a point-by-point cautious consideration.

(... ) She had a pathogenic variant of HDAC8 gene. A heterozygous pathogenic variant in FANCG gene was identified. Clinical significance of the variant in the BRCA2 gene was uncertain. They considered that contribution of HDAC8 and FA/BRCA pathway multigenic heterozygous variants might explain the clinical manifestation of this neurodevelopmental disease. It is doubtful that heterozygous pathogenic variant in FANCG causes chromosomal instability. Microdeletion of FANCG may be found by MLPA study. Or RNA seq may find other variants in FANCG or other genes.

The presence of CNVs (such as microdeletions in FANCG) were excluded either by aCGH or by exome sequencing data analyses. Furthermore, a microdeletion in FANCG is very improbable, as a good sequencing coverage was attained. In case of a microdeletion leading to no protein expression, a severe Fanconi anemia phenotype together with chromosomal instability would be expected.

Nevertheless, the authors are studying this case by biochemical assays and live microscopy. To dissect the importance of each variant towards the phenotypes observed in the patient cells we will work on HDAC8 knockout cells transfected with plasmidic versions of the variants separately or in combination. We will also study probands fibroblasts to see the capacity to form RAD51 foci when challenged with Rad51 to see the proficiency of the BRCA2 gene in the proband. We will also challenge the proband’s cells in order to see FANCD2-S and FANCD2-L to confirm that the FA/BRCA pathway is not with a normal performance in the proband.

With this paper we wanted to highlight the clinical manifestation caused by the three variants to explain this NDD case. This paper did not intend to be biochemical or to clarify this, but instead warrant future clinical and biochemical investigations on these matters.

The authors thank the reviewer’s comments, and provided their inputs and agree with this revised version of the manuscript. We trust the current version meets with your approval, with that of the reviewers as well as adhering to the journals’ formatting requirements.

Thanking in advance.

Yours sincerely,

Paula Jorge
